# ADAPTING CONVNETS FOR NEW CAMERAS WITHOUT RETRAINING

## ABSTRACT

In the vast majority of research, it is assumed images will be perspective or can be rectified to a perspective projection. However, in many applications it is beneficial to use non conventional cameras, such as fisheye cameras, that have a larger field of view (FOV). The issue arises that these large FOV images can't be rectified to a perspective projection without significant cropping of the original image. To address this issue we propose Rectify Convolutions (RectConv); a new approach for adapting pre-trained convolutional networks to operate with new non-perspective images, without any retraining. Replacing the convolutional layers of the network with RectConv layers allows the network to see both rectified patches and the entire FOV. We demonstrate RectConv adapting multiple pre-trained networks to perform segmentation and detection on fisheye imagery from two publicly available datasets. Our method shows improved results over both direct application of the network and naive pre-rectification of imagery. Our approach requires no additional data or training, and we develop a software tool that transforms existing pre-trained networks to operate on new camera geometries. We believe this work is a significant step toward adapting the vast resources available for perspective images to operate across a broad range of camera geometries. Code available upon acceptance.

## 1 INTRODUCTION

Computer vision has seen massive advancement and adoption over the past few decades. This success is built largely on the power of artificial neural networks and the widespread availability of both the compute and datasets required to train them. These approaches generally require training data representative of both the intended operating environment and the cameras used to measure that environment. Consequently, adapting to advances in imaging technology generally requires gathering extensive new datasets reflective of new camera properties, even when the operating environment remains unchanged.

In this paper we propose a training-free approach to domain adaptation that modifies pre-trained neural networks to operate with previously unseen cameras. Under the assumption that the training domain remains unchanged, we adapt the neural network to interpret that domain as seen through different cameras.

We believe this approach could have applicability across a broad range of neural architectures and camera technologies. For this paper we focus on convolutional neural networks (CNNs) trained on conventional monocular imagery, and demonstrate adaptation to wide-field-of-view (FOV) fisheye-lens imagery. We show why adaptation is required, why naive approaches like image rectification fail, and that our approach delivers performance comparable to the source domain. We show this working with multiple tasks, networks, and cameras. We anticipate generalisation to new imaging geometries and neural architectures to be well within reach.

Our approach generalises one of the fundamental assumptions underlying convolutional neural networks: translational invariance. This is a product of the shared weights of the convolutional kernel used across the image, and it greatly reduces the number of parameters to learn. Imagery captured with conventional cameras mostly exhibits this translational invariance, and any distortion due to lens non-ideality can generally be calibrated and rectified away so that it doesn't impact performance. However unconventional camera geometries including fisheye-lens cameras do not exhibit

this sort of invariance, and so a part of a scene captured by a fisheye lens might look quite different depending where it is in the image. This means that applying a CNN trained on conventional imagery to a fisheye image yields poor performance.

This same observation applies across a breadth of imaging geometries for which translational invariance does not apply. A naive solution is to calibrate the camera and rectify its imagery, such that translational invariance holds. This would have the benefits of allowing convolutional layers to work well across the entire image, and across images captured by different camera geometries. However rectification is not always feasible, as is evident with large-FOV cameras for which no mapping to a rectified image is possible without cropping and losing parts of the original image (Courbon et al., 2007) (see Figure 1). Another approach would be to split the input image into rectified patches and query the network multiple times, this has a couple drawbacks. It incurs additional computation cost as multiple projections and inferences need to be computed. Also specific design decisions need to be made around the size location and amount of patches used (Su & Grauman, 2017).

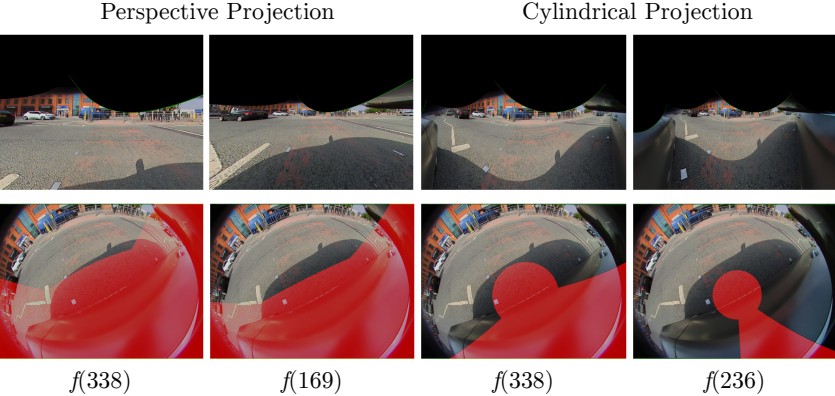

Figure 1: Example of perspective and cylindrical camera projections applied to a wide field of view fisheye image. Regions in red show areas that are excluded from the rectified projection. Decreasing the focal length can reduce cropping but increases distortion.

To overcome these limitations we introduce Rectifying Convolution (RectConv), a modified convolutional layer based on deformable convolutions (Dai et al., 2017). As depicted in Figure 2, Rect-Conv flips the paradigm that images should be warped to match the kernel shape, instead warping the kernel shape to match the local shape of the image. Replacing normal convolution layers with RectConv layers allows pre-trained networks to operate on new imaging geometries with improved performance. The only additional information required is a calibrated model of the camera, from which the RectConv deformations are computed.

The main contributions for this work are: **(1)** We propose RectConv locally rectified convolution layers and show that by performing a local layer-wise warping of the kernels within a CNN both the local and global deformations can be described and accounted for. This structure allows networks to natively deal with previously unseen camera geometries. **(2)** We develop a software tool that adapts pre-trained convolutional networks to imagery from calibrated previously unseen cameras, allowing application of networks to new domains without additional training or data. **(3)** We demonstrate improved performance compared to naively applying pre-trained networks or performing rectification. We show this improved performance for wide-FOV images on multiple networks architectures, cameras and tasks.

We will make our code available for adapting pre-trained neural networks to new camera geometries. We believe this work is an important step in adapting approaches from computer vision to a broader range of existing and emerging camera technologies. While we focus on large-FOV cameras and fully convolutional neural networks, we anticipate extension to other network architectures and camera geometries is feasible.

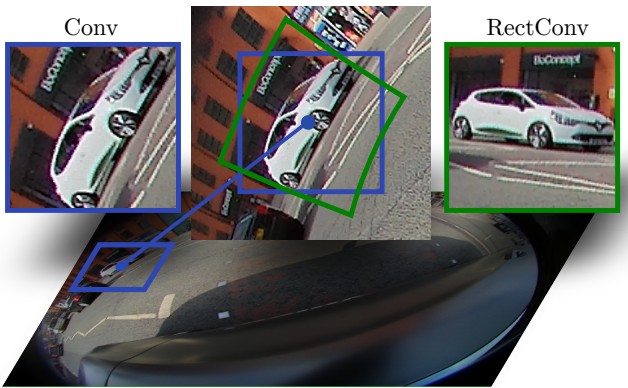

Figure 2: An example of what regular convolution and RectConv see for a fisheye image at a given position in the image. Blue and green boxes indicate the kernel shapes for regular convolution and RectConv, respectively.

## 2 RELATED WORK

**Existing Approach for large-FOV cameras.** Much research has gone into using large-FOV images, such as fisheye (Plaut et al., 2021; Rashed et al., 2021) and panorama (Yang et al., 2018; 2019). These larger fields of views can be hugely beneficial for specific applications like autonomous driving (Ye et al., 2020). One common technique is to transform an existing perspective dataset to look like a large-FOV image e.g. fisheye (Kim & Park, 2022), to aid in the training process. This allows existing datasets of conventional perspective images to be used, but requires retraining per camera geometry and does not entirely capture the target domain behaviour.

While there is a trend toward transformer-based architectures for many applications, CNN-based methods remain state of the art for fisheye many applications (Deng et al., 2017; Sáez et al., 2018; Ye et al., 2020). These approaches require extensive datasets and training for the specific type of camera being used. This can be prohibitive, fails to benefit from the extensive existing pre-trained networks and datasets, and limits generalisation to different cameras.

**Adapting convolutions.** There are multiple works that aim to adapt convolutional layers to better suit a specific camera or purpose such as video (Huang et al., 2022). Jaderberg et al. (2015) were among the first to adapt convolutions to learn spatial tranformations, with their Spatial Transform Networks (STNs). STN address global transformations but fail to compensate for local deformation. Follow-on work by Jeon & Kim (2017) proposed Active Convolutions and applied a learnt offset for sampling locations, however this work only applied a single offset across the whole image, again failing to address local deformation.

Deformable Convolutions (Dai et al., 2017; Zhu et al., 2019) are a more general version of active convolutions which learn an offset field mapping position in the image to local deformation. This makes it much more general at the cost of an increased number of learned parameters. Our work builds directly upon deformable convolutions with one key difference: instead of using deformable convolutions during training we employ camera calibration to derive a closed-form offset field to match the geometry of the input imagery. Our goal is fundamentally different from the original work in that we are aiming to adapt existing networks to new camera geometries, and not aiming to improve performance on imagery already in the training data.

**Spherical convolutions.** The line of work which is most closely related to ours is spherical convolutions (Su & Grauman, 2017) and the many follow on works (Coors et al., 2018; Esteves et al., 2018). This work applies CNNs trained on perspective images to 360° images. While these approaches require additional training or fine tuning, our goal is to adapt to new cameras without retraining. Su & Grauman (2019) introduces Kernel Transformer Networks with a similar goal to ours in that they seek efficient adaptation of existing models from perspective imagery without retraining. Their approach is specific to 360° imagery and does not generalise well to other camera geometries. We introduce a methdolology that can adapt networks to a broad range of imaging geometries, which we demonstrate on fisheye images that are not well addressed by spherical models (Yogamani et al., 2019).

## 3 RECTIFY CONVOLUTIONS

### 3.1 RECTCONV LAYERS

To achieve the outlined goals we propose an adaptation of the convolutional layer which we call RectConv. A RectConv layer does not have a fixed kernel shape, instead the shape matches the local deformation at the point in the image that the kernel is being applied to. This local deformation results in kernel "offsets' which are calculated based on how the patch would be rectified. Figure 2 shows an example of the RectConv kernel shape and the corresponding view observed from that kernel. This adaptation of the convolutional layer is based on deformable convolutions, which provide a general framework for warping kernel shapes for each pixel location in an image. To achieve this, we require a way to calculate the local kernel offsets required for each pixel based on calibrated camera parameters. The offsets also need to be adjusted for each different network layer, especially for layers that modify size, such as pooling.

**Camera Model.** Our method requires an invertible camera model that projects image points into 3D points of intersection with some reference surface in 3D space. A general form for such a model is $p = f_{3D}(u, v)$, where $p$ is a point in space in Cartesian coordinates $(x, y, z)$ and $u, v$ are coordinates on the image plane. The function $f_{3D}$ is the camera projection from 2D to 3D. The inverse form is also required, $u, v = f_{2D}(p)$, where $f_{2D}$ converts a point in 3D space to its corresponding 2D image coordinates. Models of the required form are readily available for a broad range of cameras including the fisheye-lens cameras used in this work.

**Calculating Kernel Offsets.** Here we derive the kernel offsets required at each image location. The process is depicted graphically in Figure 3. The first step is to convert pixels to points of intersection with a reference surface in 3D space,

$$p_i = f_{3D}(u_i, v_i), \tag{1}$$

where $i$ denotes the different positions in the kernel. E.g. for a $3 \times 3$ kernel size this would be calculated nine times. The scale of the patch in 3D space is computed as

$$s = \frac{w_{grid} + h_{grid}}{2}, \tag{2}$$

where $w_{grid}, h_{grid}$ are the horizontal and vertical size of the original grid and are calculated as $p_{max} - p_{min}$ in their respective dimensions. $s$ denotes the scale of the averaged width and height and is required in scaling a regular resampling grid. A linear planar sampling $k$ of scale $s$ is calculated to be tangential to the point $p_c$ at the centre of the original grid, from which new sample points can be calculated as

$$\hat{p}_i = p_c + k_i. \tag{3}$$

Here $\hat{p}_i$ is the new point in space at position $i$ in the kernel. With the new list of points in 3D space that can be converted back to the image plane,

$$\hat{u}_i, \hat{v}_i = f_{2D}(\hat{p}_i), \tag{4}$$

where $\hat{u}_i, \hat{v}_i$ are the rectified pixel location on the image that the convolution should sample. For use within the deformable convolution framework these points are converted to offset as $offset_i = \hat{u}_i - u_i, \hat{v}_i - v_i$. This offset value needs to be calculated for every position in the kernel and every position in the image. To reduce computation time we compute offsets for a subset of image locations and interpolate. As the cameras used had a continuous smooth projection, is effectively reduces computation time without affecting performance.

**Modifying Offsets.** Different types of CNN layer impact output size and scaling in different ways, and the kernel offset field needs to account for these variations in size. A layer with no padding results in a truncated output that can be accommodated by cropping the offset field. A layer with stride can be accommodated by down-sampling the offset field at the appropriate rate. Dilation results in an effective increase in kernel size, so an offset of a larger kernel should be used then sampled in the same way as the dilation process. Finally, convolutional layers after a dimensional reduction, such as from a pooling layer or a previous convolution with stride, are accommodated by scaling the offset field in both size and magnitude. As the input size is scaled the offset values should be scaled by the same amount.

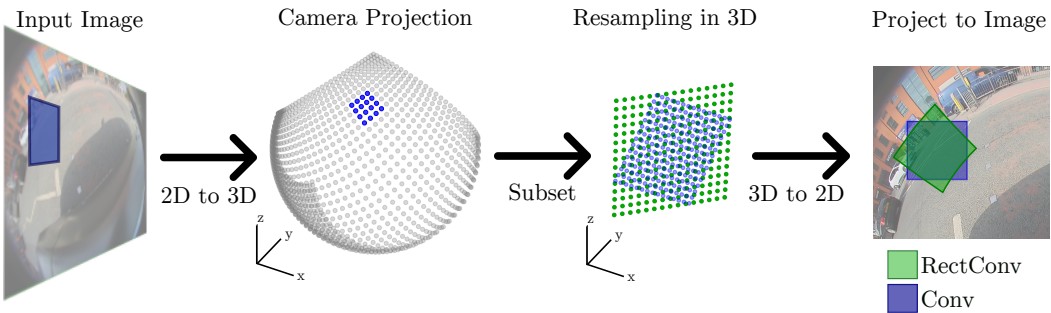

Figure 3: For a given patch each pixel is converted to 3D space which is then sampled on a regular planar grid. This grid in 3D space is converted back to image locations that represent the kernel locations for that position.

**Conversion from Conv to RectConv Layers.** Conversion from a conventional CNN network to a RectConv version can be carried out efficiently and elegantly. Given a pre-trained model and camera parameters, a recursive search through the network modules identifies all the convolutional layers and replaces them with a RectConv layer. Offsets for the RectConv are computed as in the previous section, and weights and bias terms from the pretrained network are left unmodified. All convolution layers with a kernel size greater than one are converted to a RectConv layer in this manner. Layers with a kernel size of one require no modification and are left unchanged.

### 3.2 EFFECTS OF INTERPOLATION.

A consequence of employing non-integer offset fields in the deformable convolutions is that samples must be interpolated from the input imagery. Our implementation employs bilinear interpolation (Dai et al., 2017). This process is information destroying and is present in every RectConv layer. The slight error at each layer accumulates as it propagates through the network.

Figure 4 illustrates the effect of interpolation on the output of a network. This experiment shows a histogram of a binary classification network's output before a final classification layer is applied. The network used for this demonstration was a simple CNN, which has 4 convolutional layers with kernel sizes of 7, 5, 5 and 3, with no padding or stride. Then 4 additional $1 \times 1$ convolutional layers. The network was trained on a binary cats and dogs dataset. The figure compares the convolutional form of the network applied to rectified perspective imagery, and the RectConv version applied to a distorted version of the same images. For an ideal conversion be-

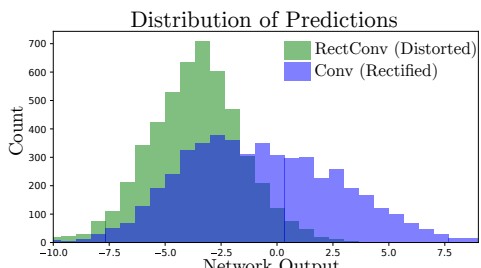

Figure 4: A histogram of the outputs from a binary classification task showing how a RectConv layers result in a bias shift in the outputs.

tween convolutional and RectConv networks the outputs would be identical. However a shift in the distribution is evident, and we hypothesize arises due to the compounded impact of interpolation in the RectConv approach.

While these results show RectConv conversion is imperfect, it nevertheless demonstrates competitive performance in adapting to new camera geometries without a need for retraining. We leave further exploration and mitigation of the impact of interpolation as future work.

**Model Architecture.** While RectConv layers could theoretically be applied to any CNN, there is some criteria that should be met. The chosen network shouldn't have a fixed input size and aspect ratio. As the target camera is not likely to be the same size as the trained data, the input images would need to be modified to even be accepted by the network, these networks usually have fully connected layers which require a fixed size input.

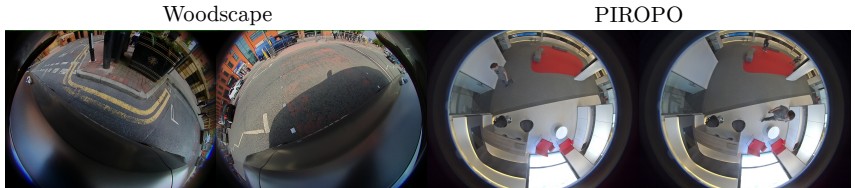

Figure 5: Example images from each data are shown. Both have wide-FOV images where full image rectification is not possible

From this we chose to develop this approach using fully convolutional networks (Long et al., 2015; Dai et al., 2016). These CNN architectures can be used for variety of different computer vision tasks. These networks have do not have any fully connected layers and natively accept images of different sizes. These networks decoders are generally simple using some form of upscaling. As such deconvolutional layers is component that, at the time of writing, we haven't explicitly dealt with. That is not to say that they cannot also be adapted to a rectified alternative, similar to convolutional layers, just that for our work it wasn't required, so have been left as future work.

## 4 EXPERIMENTS

**Tasks.** We believe the proposed approach is general and applicable across many vision tasks. There are however certain tasks which are better suited to RectConv conversion than others. Size-conserving and pixel-wise labelling tasks such as segmentation and depth estimation are a strong fit, and for this reason we chose segmentation to demonstrate the effectiveness of the approach.

More challenging tasks have outputs with different dimension to the input. An key example is object detection for which the outputs are a list of bounding boxes with pixel locations. We chose this task as a more challenging example for RectConv networks. An interesting side effect of conversion to RectConv layers is that because the kernels only see rectified patches the bounding boxe extents have been rectified locally around the object being bounded. This necessitates an additional step in which the rectified box extents are projected back to the original input image.

**Datasets.** We demonstrated our approach on imagery from three different cameras drawn from two separate datasets. Firstly, Woodscape (Yogamani et al., 2019) is a multi-task, multi-camera fisheye dataset. Woodscape has four fisheye cameras deployed on a vehicle, with data collected throughout a city environment. While Woodscape has multiple tasks, we focus on segmentation which has ground truth labels for 9 classes. We demonstrated results using two cameras which capture the diversity of imaging geometries present in the dataset.

The second dataset we use is PIROPO (del Blanco et al., 2021). This dataset tracks people moving around a room from both a perspective and omnidirectional camera across multiple sequences. We used only the omnidirctional camera and demonstrated for both segmentation and detection. The ground truth data provided includes a single labelled point for each person.Typical images from each dataset are seen in Figures 5.

**Models.** We demonstrated our approach adapting to four different pre-trained segmentation models constructed from two different backbones, ResNet50 and ResNet101 (He et al., 2016), and three separate architectures, FCN (Long et al., 2015), DeepLabV3 and DeepLabV3+ (Chen et al., 2017). We used an FCOS ResNet50 (Tian et al., 2019) for object detection. These are representative of standard models for segmentation and detection, while also having readily available pre-trained weights.

**Pre-trained Networks.** Each test required a pre-trained convolutional network to be converted to the RectConv version. Depending on the dataset used and the corresponding required labels, we used networks pre-trained either on Cityscape (Cordts et al., 2016) or Pascal VOC Everingham et al. (2010), which contain only imagery captured using conventional cameras. For the Woodscape dataset all pre-trained models were trained on the Cityscape dataset. Segmentation was evaluated using only the classes present in both Cityscape and Woodscape. In the case of DeeplabV3+ we use a publicly available network pre-trained on Cityscape. For the PIROPO dataset all pre-trained

Table 1: Comparison of segmentation results for different pre-trained models applied to fisheye imagery from the Woodscape dataset, with and without rectifying convolutions. Bold denotes the best result. Showing the mean intersection over union (MIOU) and the pixel accuracy

| Camera | Method | FCN(Resnet50) | | FCN(Resnet101) | | DeeplabV3(Resnet50) | | DeeplabV3+(Resnet101) | |
|---|---|---|---|---|---|---|---|---|---|
| | | Pixel Acc | MIOU | Pixel Acc | MIOU | Pixel Acc | MIOU | Pixel Acc | MIOU |
| | Conv | 83.16 | 24.90 | 82.60 | 24.90 | 73.64 | 22.68 | 81.03 | 22.76 |
| Camera 1 | Conv(Rectified) | 82.05 | 27.01 | 85.27 | 28.13 | 72.64 | 19.38 | 86.45 | 29.02 |
| | RectConv(Ours) | **86.83** | **29.50** | **88.17** | **30.02** | **80.6** | **26.19** | **88.72** | **29.84** |
| | Conv | 84.64 | 24.09 | 84.97 | 24.53 | 73.12 | 20.74 | 79.77 | 21.31 |
| Camera 2 | Conv(Rectified) | 83.56 | 24.88 | 85.14 | 25.62 | 77.81 | 21.28 | 86.21 | **26.37** |
| | RectConv(Ours) | **87.04** | **26.76** | **87.87** | **27.38** | **77.89** | **22.71** | **86.55** | 25.70 |

| Ground Truth | Conv | Rectifing | RectConv |
|---|---|---|---|

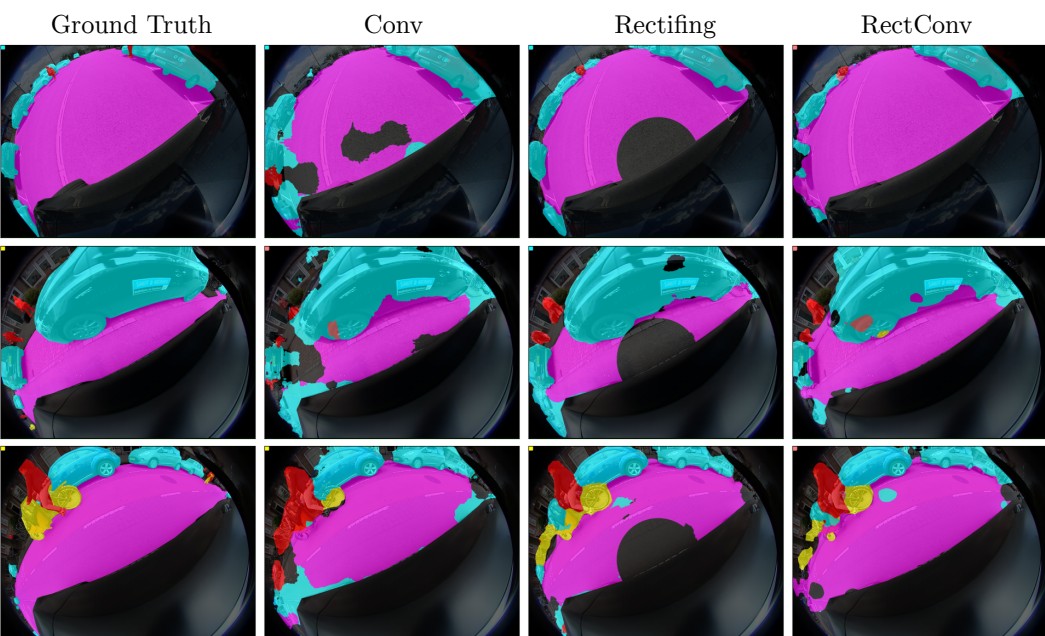

Figure 6: Comparison of segmentation using a FCN-Resnet101 pre-trained on Cityscape. The un-modified pre-trained network shows poor performance, pre-rectification shows poor performance and suffers from dead zones that could not be included in the rectification, and the proposed Rect-Conv shows the strongest performance while also covering the entire image.

models were trained on Pascal VOC, we used the pre-trained models supplied by pytorch which are readily available. For this test only the person class was used and the other available classes were ignored.

## 4.1 RESULTS

To provide evidence for the generality of the proposed approach, we evaluate performance across a range of models, cameras, datasets and tasks.

**Segmentation.** We evaluated Segmentation performance across both datasets. Table 1 and Figure 6 show quantitative and qualitative results for the Woodscape dataset. We compare our method to the naive approach of applying the pre-trained network directly to the distorted image, as well as pre-rectifying the input images before inference. When rectifying we use a cylindrical projection as this projection maintains a larger field of view compared to other projection. Our method outperforms the alternatives with stronger qualitative results that cover the entire FOV, as well as stronger quantitative results in all metrics except one. Importantly, our approach requires no additional training, only a one-time closed-form conversion of the convolutional layers to their RectConv alternatives.

Segmentation                    Object Detection

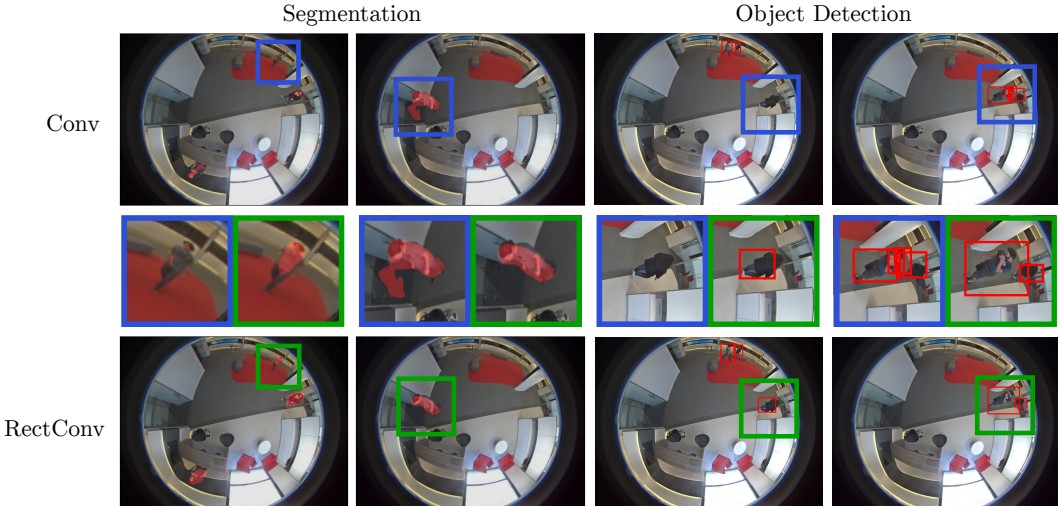

Figure 7: Detection and segmentation results for people on the PIROPO dataset (del Blanco et al., 2021) using pre-trained segmentation (FCN-Resnet101) and object detection (FCOS-Resnet50) networks. Converting to RectConv layers has improved the segmentation and object detection results in both removing false detection while also improving true detections. In the case of object detection we can see that the RectConv method bounding boxes have errors.

Table 2: Comparison of segmentation and detection using different pre-trained models on fisheye imagery from the PIROPO dataset.

| Method | Segmentation Network | | | | | | Ojebect Detection Network | | |
|---|---|---|---|---|---|---|---|---|---|
| | FCN(Resnet101) | | | DeeplabV3(Resnet101) | | | FCOS(ResNet50) | | |
| | Precision | Recall | F1 Score | Precision | Recall | F1 Score | Precision | Recall | F1 Score |
| Conv | 69.09 | 32.90 | 44.57 | 64.10 | 28.86 | 39.80 | 84.64 | 63.64 | 72.65 |
| RectConv (ours) | **76.91** | **48.05** | **59.15** | **81.00** | **44.30** | **57.28** | **86.67** | **65.66** | **74.71** |

Figure 6 clearly illustrates the differences between approaches and how they compare to the ground truth. For the pre-rectification method the results are distorted back to the original image geometry, making the cropped dead zone clearly visible in the centre of each segmentation mask. The additional errors from using a distorted input are also clearly visible for the convolution method, especially looking at the road prediction.

Segmentation results for PIROPO are shown in Table 2 and Figure 7. As there was no ground truth segmentation masks for this dataset the quantitative results are shown as an accuracy of correct detection of people within the image. These results show that not only does converting layers to RectConv increase true detection, it also reduces spurious detection. From the qualitative results we can see that the actual segmentation masks are also cleaner compared to the naive approach.

**Object Detection.**

Object detection results for PIROPO are shown in Table 2 and Figure 7. RectConv outperforms the conventional approach in all measures, with clear advantages in both segmentation and detection. The extent of the quantitative benefit is not as strong in detection as it is in segmentation. Conversion of bounding boxes in the RectConv network into image space is imperfect and is a topic for future research. Typical errors from the bounding box conversion are seen in Figure 7. Another issue impacting performance is that the RectConv version of the FCOS network is prone to double detection, which we attribute to the kernel interpolation process resulting in spreading of detection peaks. We anticipate addressing these issues in future work, noting that overall detection and segmentation performance are both nevertheless improved by the RectConv approach.

Table 3: Comparison of inference times using the rectifing convolutions

| Model | Inference Time (s) | | Change (%) |
|---|---|---|---|
| | Conv | RectConv | |
| FCN Resnet50 | 0.30 | 0.46 | 53.13 |
| FCN Resnet101 | 0.41 | 0.66 | 60.23 |
| DeeplabV3 Resnet50 | 0.32 | 0.55 | 71.01 |
| DeeplabV3+ Resnet101 | 0.25 | 0.38 | 52.83 |

Table 4: Effect of different networks parts converted of RectifyConv layers

| RectConv Backbone | RectConv Classification Head | Pixel Acc | MIOU |
|---|---|---|---|
| ✗ | ✗ | 82.60 | 24.90 |
| ✓ | ✗ | 88.14 | 30.01 |
| ✗ | ✓ | 82.63 | 24.92 |
| ✓ | ✓ | **88.17** | **30.02** |

**Inference Time** The conversion of a network to use RectConv layers incurs additional inference-time computational cost due to the additional step of deforming the kernels. Table 3 shows the average inference time for the four nextworks we evaluated operating on a single image, running on an NVIDIA RTX3060, and the percentage increase in time. Across the four different models there was an average of 60% increase in time.

## 4.2 Ablation Study

Table 4 shows the results of an ablation study on how converting different parts of the network to RectConv layers affects overall performance. This study was performed on the Woodscape dataset, using the FCN ResNet101 segmentation network. We can see that while converting both backbone and classification heads yields performance improvements, most of the performance gain comes from converting the backbone. As the backbone is the part that is extracting geometric features it makes sense that it is the part that benefits most from RectConv. This lends credit to this method being about to be adapted for other networks and tasks as these backbones are fairly universal.

## 5 Conclusions

We introduced RectConv, a training-free method for adapting pre-trained networks to imagery captured with previously unseen cameras. We achieved this by generalising the translational invariance assumption underlying convolutional neural networks, introducing a spatially-varying deformation that allows existing networks to effectively operate over imagery captured from a broad range of imaging geometries.

We demonstrated our approach adapting segmentation and detection networks pre-trained on conventional imagery to operate on fisheye imagery from three cameras drawn from two publicly available datasets. RectConv outperforms direct application of pre-trained networks and naive rectification, and unlike rectification RectConv maintains a full field of view.

We believe this work is a significant step toward generalising neural network models to operate across a broad range of cameras without the need for additional datasets or retraining. In future we aim to demonstrate RectConv on additional tasks such as depth and pose estimation, with additional camera geometries, and with a broader range of network architectures. There is scope to expand the network conversion process to handle additional layer types like deconvolution, and as discussed in the Results section to improve performance around bounding box conversion and multiple detections.

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
