# OpenReview forum: "Adapting ConvNets for New Cameras without Retraining"
_ICLR.cc/2024/Conference — ICLR 2024 Conference Withdrawn Submission_

### Official Review · Reviewer_v5En · 2023-10-31

**Soundness:** 3 good
**Presentation:** 3 good
**Contribution:** 2 fair
**Rating:** 5
**Confidence:** 5

**Summary:**

This work proposes a training-free approach to domain adaptation that modifies pre-trained neural networks to operate with previously unseen cameras. In particular, the authors reformulate the sampling strategy of vanilla convolutions in terms of the camera imaging process, given a calibrated camera model. Experimental results show this work is able to improve performance for wide-FOV images on multiple network architectures, cameras, and tasks.

**Strengths:**

+ The proposed work shows the potential to adapt approaches from existing learning models to a broader range of camera models.
+ A new training-free strategy is designed to achieve adapting convnets for new cameras.
+ Compared to the existing deformable convolutions, the proposed work employs the practical perspective of camera calibration to derive a closed-form offset field to match the geometry of the input imagery.

**Weaknesses:**

- Some vital details regarding the proposed RectConv layers are missing. For example, how do the padding, down-sampling, dilation operations change when modifying offsets based on different calibrated camera models? More details and concrete formulations are expected.
- The application and generalization of this work seem limited because the proposed RectConv requires a calibrated camera model. However, lots of cases are blind to the camera model and camera parameters, such as the images derived from the Internet and in the wild.
- The comparison on the setting of Conv (Rectified) is unfair. The proposed RectConv uses the accurately calibrated camera parameters to perceive the radial distortion in the fisheye image. However, the authors simply rectify the fisheye image using a cylindrical projection, which cannot well eliminate the distortions (as shown in Figure 6). A fair comparison could be achieved using the calibrated camera parameter to rectify the fisheye image and insure there are no significant distortions.
- Once we have the calibrated camera model and its parameters, why not directly calibrate the fisheye/wide-angle uncalibrated image? Such an implementation also does not require retraining the downstream vision perception models. More clarifications are expected to be provided.
- Since the key step of this work is to use the way of camera calibration to reformulate the sampling strategy of vanilla convolutions, the reviews on camera calibration methods are needed. For example,  "A perceptual measure for deep single image camera calibration", "DR-GAN: Automatic Radial Distortion Rectification Using Conditional GAN in Real-Time", "DeepCalib: a deep learning approach for automatic intrinsic calibration of wide field-of-view cameras", "Deep single image camera calibration with radial distortion", "Deep Learning for Camera Calibration and Beyond: A Survey", etc. The authors are suggested to provide some discussions on the above literature and explain how the development of camera calibration motivates this work.
- The details of the experimental implementations are missing. For example, what's the specific camera model and camera parameters using in this work? How's the performance on evaluations of the cross-camera model?

**Questions:**

Could this work extend the camera model from a parametric model to a non-parametric model? More discussions would be helpful for future explorations.

---

### Official Review · Reviewer_Vp9L · 2023-10-31

**Soundness:** 3 good
**Presentation:** 3 good
**Contribution:** 1 poor
**Rating:** 3
**Confidence:** 5

**Summary:**

The authors propose a method to adapt a network trained on perspective projected standard images to work natively on distorted images acquired by a variety of cameras (with a particular focus on fisheye cameras). They propose to do so for convolutional networks without retraining them but simply distorting the sampling grid of convolutional layers according to the camera distortion model. This proves to be effective when evaluating different models on two fisheye datasets for semantic segmentation and object detection. Comparisons are carried out against running the standard convolutional models on distorted images and running on a rectified (but cropped) version of the images. The distortion aware convolutional model achieves consistently better performance.

**Strengths:**

+ The method is undoubtedly useful for many practical applications and can be applied as it is on many off the shelf models with some degree of success.

**Weaknesses:**

a. **No novelty & lack of related works**: The paper failed to mention a crucial related work [Distortion-Aware Convolutional Filters for Dense Prediction in Panoramic Images](https://www.ecva.net/papers/eccv_2018/papers_ECCV/papers/Keisuke_Tateno_Distortion-Aware_Convolutional_Filters_ECCV_2018_paper.pdf) which, it seems to me, already proposed all the contributions of this work in 2018. As such this work has basically no novelty except “rediscovering” the same idea and running tests on newer datasets and tasks.
Other works which are relevant and not considered: [Bending Reality: Distortion-Aware Transformers for Adapting to Panoramic Semantic Segmentation](https://openaccess.thecvf.com/content/CVPR2022/html/Zhang_Bending_Reality_Distortion-Aware_Transformers_for_Adapting_to_Panoramic_Semantic_Segmentation_CVPR_2022_paper.html) and [CAM-Convs: Camera-Aware Multi-Scale Convolutions for Single-View Depth](https://openaccess.thecvf.com/content_CVPR_2019/papers/Facil_CAM-Convs_Camera-Aware_Multi-Scale_Convolutions_for_Single-View_Depth_CVPR_2019_paper.pdf). All these related works should have been included in the discussion of the paper and I would have expected a comparison to them. The focus of these works is mostly on panoramic images but many of the problems highlighted by athe authors are the same.

b. **Significant impact on runtime for modest improvement in performance**: The proposed modified convolutional operator according to Tab. 3 increases the runtime by up to 50% for an average gain in quality of few percent (see Tab. 1). This is a huge trade-off that makes the method impractical for many applications. Also Sec. 3.2 highlights how the deformed convolutional operator actually propagates several errors in the network and drastically changes the distribution of values predicted. Therefore the claim of the authors that the proposed layer does not modify the behavior of a trained network is only partially true.

c. **Missing baselines**: I would have liked the method to be compared to a version of the images patchified and rectified for each patch to minimize the space being cropped due to the rectification procedure used.

**Questions:**

* Can you comment on weakness (a) and in case my understanding is wrong explain what is the main difference between your proposal and Tateno 2018?

* Why is Tab. 2 not reporting results on rectified images?

* Have you tried to implement a version of the model that does not use bilinear sampling but just NN-sampling? Would it speed up the inference significantly? What would be the impact on performance?

**Details Of Ethics Concerns:**

No concern

---

### Official Review · Reviewer_Qoap · 2023-11-01

**Soundness:** 2 fair
**Presentation:** 3 good
**Contribution:** 3 good
**Rating:** 5
**Confidence:** 4

**Summary:**

- Authors propose a training free approach to employ pre-trained CNNs to cameras with large Field of View (FOV). The motivation of the work is to leverage large number of pre-trained models trained on visual data captured from traditional perspective cameras without resorting to approaches like image rectification.
- This approach is in contrast to earlier work which generally rectify and warp the input image (either perspective images to fisheye to aid during training or fisheye to perspective image patches).
- The method borrows the methodology from line of work that specialize deformable convolutions to perform a more specialized convolution (e.g. NeurVPS [1], for instance, derives an offset field that tries to match lines emanating from a candidate vanishing point), in their case, the algorithm performs a local layer-wise warping of the kernels. They do so by deriving a closed form offset field to match the geometry of the input imagery.

[1] NeurVPS: Neural Vanishing Point Scanning via Conic Convolution, NeurIPS 2019

**Strengths:**

- RectConv is a novel method. I really liked the motivation of the paper and also the methodology and the proposed algorithm to leverage existing pre-trained CNN's.
- Potential Impact of this method is high. The method can be used to adapt any fully convolutional network without requiring retraining.
- The method is shown to improve over pre-trained segmentation and detection models. The qualitative results in Fig 6 are helpful.

**Weaknesses:**

- I have some concerns with evaluation and feel that it's not complete.
- Are there any limitations/bottleneck while finetuning RectConv network on the target fisheye data? As there is some training data available in Woodscape and PIROPO data, it makes sense to also observe results when a limited amount of that data is used (finetuning both the baseline CNN and RectConv-adapted CNN).
    -  For instance, looking at Table 4, it appears that maybe finetuning the backbone with RectConv might help object detection.
- While the paper discusses the effect of interpolation, the paper does not present an in depth analysis of the effect. Maybe fine-tuning the network would alleviate this issue?
- While Table 3 presents results on runtime (and a 60% increase is quite large), is there an estimate of the number of FLOPs? Quite possibly, there might be room for improvements in the implementation. A discussion would be appreciated.
- Needs proofreading. The language of the paper is not clear at a few places (E.g. Model Architecture paragraph).
- Please fix typographical errors: "Ojebect Detection Network" in Table 2.

**Questions:**

Please answer the questions in Weaknesses apart from the following questions.
- I have a question about Table 2, specifically the Object Detection performance. What is the IOU threshold of the metric employed? I'm assume Precision corresponds to AP and Recall corresponds to AR respectively, please clarify.
- Both the datasets used are captured from fisheye cameras. The paper makes the claim that the method ought to generalize to other cameras, could the authors substantiate that claim?

---

### Official Review · Reviewer_v97K · 2023-11-02

**Soundness:** 2 fair
**Presentation:** 2 fair
**Contribution:** 2 fair
**Rating:** 1
**Confidence:** 4

**Summary:**

This paper proposes to deform the convolution kernel during forwarding instead of undistorting the image as preprocessing. To achieve this, the authors undistort the local coordinate map centered at each pixel, resample the pixels, and then apply convolution to the resampled local region. To reduce the extensive undistortion and sampling computation, the authors also make efforts to accelerate the computing, such as interpolating the coordinate map from coarse grids. To demonstrate the effectiveness of the proposed method, they conduct experiments on multiple tasks using the PIROPO and Woodscape datasets and achieve better performance than baselines that undistort as preprocessing and then apply fixed-kernel convolution.

**Strengths:**

- The motivation is well introduced.
- The experiment is comprehensive, especially the distribution shift visualization.

**Weaknesses:**

- The core of the problem is that there is no single-pass transformation that can make an equivariant 2D image for the 3D world, e.g., both perspective and spherical images are distorted. So a popular choice is to project the image onto the tangent plane of a spherical point, i.e., undistorting spherical images into perspective local patches. This paper does not have a fundamental difference from this, but it only has dense per-pixel undistortion.
- In Sec. 2, the authors mentioned that Su & Grauman (2019) only applies to 360 (equirectangular) images, which is not true since any camera can be projected onto an equirectangular map. Moreover, Su & Grauman (2019) extract features from perspective local patches first and then project the patch features onto an equirectangular map, which should minimize the degradation of performance.
- There is an important missing reference: "Spherical CNN", ICLR '18, which solves the problem more fundamentally through group convolution. It might have inferior performance on images due to the FFT acceleration, but at least a discussion is necessary.

**Questions:**

What will be the performance like if the model is learned on perspective image but still apply the rectconv proposed in the paper?